# Shaping a low-carbon future: Uncovering the spatial-temporal effect of population aging on carbon emissions in China

Zhuoqun Li[1], Haohao Li[2]*, Xueqiang Ji[1], Yuesong Zhang[1]

1 School of Public Administration and Policy, Renmin University of China, Beijing, China, 2 School of population and health, Renmin University of China, Beijing, China

* lihaohao@ruc.edu.cn

## Abstract

### Background

With the accelerated development of the aging trend in Chinese society, the aging problem has become one of the key factors affecting sustainable economic and social development. Given the importance of controlling carbon emissions for achieving global climate goals and China's economic transformation, studying the spatial and temporal effects of population aging on carbon emissions and their pathways of action is of great significance for formulating low-carbon development strategies adapted to an aging society.

### Objective

This paper aims to explore the spatial-temporal effects of population aging on carbon emissions, identify the key pathways through which aging affects carbon emissions, and further explore the variability of these effects across different regions. The findings will provide theoretical support and empirical evidence for government departments to formulate policies to promote the coordinated development of a low-carbon society and an aging society.

### Methods

Based on the panel data of 30 provinces in China from 2004 to 2022, this paper systematically investigates the impact of population aging on carbon emission intensity from both spatial and temporal dimensions by using the spatial Durbin model and the mediating effect model. The direct effect of aging on carbon emission intensity, the spatial spillover effect, and the indirect effect through mediating variables such as residents' consumption, environmental regulation, and new urbanization are analyzed in depth.

### Results

The study found that population aging in China has significant spatial and temporal effects on carbon emissions. From the spatial dimension, there is a significant spatial spillover effect of the effect of aging on carbon emissions, and aging reduces local carbon

**Data Availability Statement:** The data in the article are all from third parties, as specified in the article, and all of these data can be accessed through the website, some of which can be obtained for a fee. All the data in the article comes from the Office for

National Bureau of Statistics of China (http://www.stats.gov.cn/).

**Funding:** This work was Supported by the National Natural Science Foundation of China under Grant No. 72274207, "Research on Cross-domain Governance Models of Beijing-Tianjin-Hebei Housing Market in the Context of Regional Synergistic Development". The funders had no role in study design, data collection and analysis, decision to publish, or preparation of the manuscript.

**Competing interests:** The authors have declared that no competing interests exist.

emissions but increases carbon emissions in adjacent regions. From the time dimension, the effect of aging on carbon emissions shows a stage characteristic, initially it will reduce carbon emissions, but with the deepening of aging, its effect may tend to weaken. In addition, this study identifies a number of key pathways through which aging affects carbon emissions, including reducing residential consumption, promoting new urbanization, and increasing the intensity of environmental regulations. Finally, this study explores the regional heterogeneity of the impact of aging on carbon emissions and its mechanism of action.

## Conclusion

This study is instructive: first, the complex impact of population aging on carbon emissions should be fully recognized to formulate a comprehensive low-carbon development strategy; second, attention should be paid to the spatial spillover effect of aging on carbon emissions to strengthen inter-regional cooperation and coordination; and lastly, differentiated low-carbon policies should be formulated to address the characteristics of aging in different regions and stages in order to promote the synergistic development of a low-carbon society and an aging society.

## 1. Introduction

Accelerated population aging and climate change due to carbon emissions are two current global challenges that cannot be ignored. With global fossil fuel consumption and energy emissions hitting record highs in 2023, causing average global temperatures to rise by nearly 1.5°C, making it the hottest year on record on the planet, the impacts of climate change are being felt acutely in all countries [1]. In the aforementioned context, achieving the ambitious temperature control targets outlined in the 2015 Paris Agreement—specifically, limiting the global average temperature increase to well below 2°C above pre-industrial levels and pursuing efforts to limit the temperature increase to 1.5°C, if feasible—has become a particularly daunting endeavor. As the world's most populous developing country, China's carbon emissions have continued to rise along with its economic development, and it is now the world's largest carbon emitter. By 2020, China's carbon emissions have reached 9.899 billion tons, rising to 30.7% of total global carbon emissions [2]. To realize the green transformation, the Chinese government has established the goal of "double carbon" (peak carbon and carbon neutrality) as one of the core tasks of national development in the new era and has committed to peak carbon emissions around 2030 [3]. Currently, China is at a critical stage of realizing its commitment to peak carbon emissions, and achieving carbon reduction is not only a positive response to international commitments but also a key strategy for promoting China's high-quality economic development.

On the other hand, along with the decline in fertility and mortality rates, the global trend of population aging is becoming more and more pronounced, and the number and proportion of elderly people are increasing in almost every country in the world. Currently, Japan is the most deeply aging country, with the proportion of its population aged 65 and above already as high as 28.7%, followed by Italy and Portugal with 23.6% and 23.1% respectively. It is expected that by 2050, the global population aged 65 and over will increase to 1.6 billion, accounting for 16% of the total population [4]. China has the largest elderly population in the world, and the pace

of aging is accelerating. By the end of 2021, China's elderly population aged 60 years and above had reached 267 million, accounting for 18.9% of the total population, and the elderly population aged 65 years and above had reached more than 200 million, accounting for 14.2% of the total population [5]. It is estimated that by 2035, the proportion of the elderly population in China will exceed 30% [6].

The relationship between population aging and carbon emissions is complex and far-reaching, and it can have direct or indirect effects on carbon emissions through changes in production activities, consumption structure, consumption scale and industrial structure. Existing literature has explored the relationship between them, but no consistent conclusion has been reached. Some studies have found that aging contributes to reducing carbon emissions, as the elderly population tends to engage in low-carbon consumption [7], meanwhile aging also promotes the upgrading of industrial structure, which is conducive to lowering carbon emissions [8]. However, other research indicates that aging may increase carbon emissions, as the products consumed by the elderly are often more resource-intensive [9], and population aging can lead to a decrease in the overall savings rate, subsequently raising per capita consumption and ultimately promoting carbon emissions [10]. Cheng et al. (2015) conducted an empirical study based on an extended stochastic STIRPAT model and found that the proportion of the population aged 60 and above has a negative impact on carbon emissions in China [11]. More recent studies have further suggested that the relationship between aging and carbon emissions may be nonlinear, exhibiting an inverted U-shaped pattern. This implies that as the degree of population aging deepens, per capita carbon emissions first increase and then decrease [12, 13]. The varying conclusions drawn from different studies could be attributed to diverse methodological approaches, study contexts, and potentially, differing degrees of aging across the samples examined. Additionally, the interplay between economic development, technological advancements, and societal preferences regarding environmental sustainability may also play significant roles in shaping the observed relationships [14].

In generally, most of the existing studies focus on the overall impact of aging on carbon emissions, but China's geographic scope is vast, and the level of economic development and population aging among provinces show significant diversity and imbalance, so there is not enough research on how aging affects the intensity of carbon emissions among regions. On the basis of existing studies, this paper incorporates the spatial and temporal effects of aging on regional carbon emission intensity into the analytical framework, and explores in depth the influence paths through multiple dimensions, such as the consumption pattern of the population, the regulation of environmental policies, and the new urbanization process.

The potential contributions of the paper are reflected in the following aspects: first, by utilizing the data at the provincial level in China from 2000 to 2022, we comprehensively analyze the impacts of population aging on regional carbon emission intensity from the dynamic perspective of temporal evolution and the static characteristics of spatial distribution, which effectively broadens the temporal and spatial dimensions of this research field. Secondly, by examining the intermediary roles of multiple pathways such as residential consumption, environmental regulation, and new urbanization, the paper provides rich empirical evidence for a deeper understanding of the complex relationship between aging and regional carbon emission intensity. Finally, this paper also conducts a regional heterogeneity analysis to reveal the differences in the impact of aging on carbon emission intensity among different regions, which provides valuable references for the government to formulate differentiated and precise carbon emission reduction policies according to the actual situation of different regions.

## 2. Theoretical analysis and research hypotheses

### 2.1 The direct impact of ageing on carbon emission

**2.1.1 Direct effects of ageing on local carbon intensity.** The mechanism of aging on carbon emission is shown in Fig 1. Individuals differ in their participation in economic activities under different life cycle conditions [15]. The participation of the elderly in socio-economic activities will also decrease due to the decline in physical function and energy [16]. Therefore, with deepening aging, the supply of labor market may decrease, thereby reducing the scale of regional production activities and reducing carbon emission to some extent [17]. The decline in the number of working-age workers brought about by the change of population structure also objectively puts forward higher requirements for the quality of labor force, thus helping to promote the training of high-quality labor force in China [18]. The improvement of the quality of the workforce will also help to further enhance the environmental awareness of the region and facilitate the implementation and popularization of low-carbon activities. At the same time, with the change of demographic dividend from quantity type to quality type, the industrial structure is forced to transform from the previous "high-emission and high-pollution" enterprises to emerging technology industries [19]. This shift has a positive impact on reducing regional carbon intensity [20]. In addition to the impact on labor supply, the impact of aging on carbon emission is also reflected in the demand side. Aging will not only affect the scale of demand for goods and services, but also affect the composition of demand [12]. In terms of energy consumption, elderly people are usually different from young people in terms of lifestyle. Elderly people tend to choose energy-saving equipment and less energy consumption methods, such as using energy-saving lamps and reducing the use of householder appliances [21]. In terms of transportation demand, compared with young people, the elderly tend to have lower demand for commuting and long-distance travel. Meanwhile, in terms of travel mode, the elderly tend to take public transportation rather than private cars. All these behaviors contribute to reducing carbon emission [8]. Based on the above analysis, this paper proposes hypothesis 1:

H1: Population ageing helps to reduce local carbon intensity

**2.1.2 Spatial effects of aging on carbon emission in neighborhoods.** The increase of the elderly population has increased the demand for quality of life and medical services in the

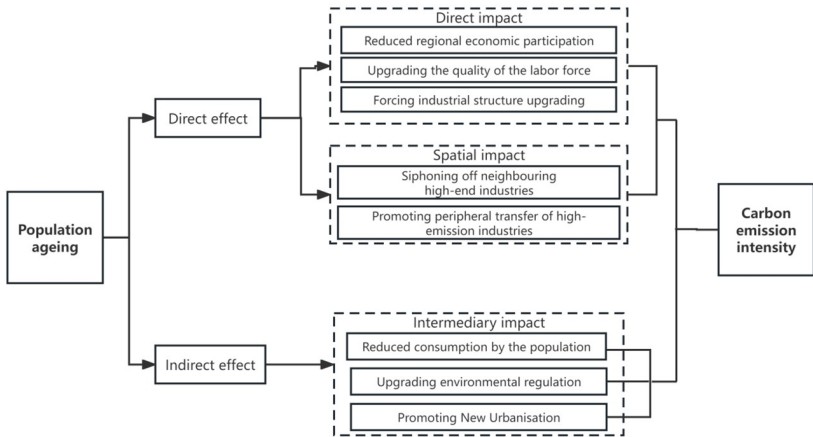

**Fig 1. Mechanism of population aging on carbon emission.**

region, and triggered the phenomenon of high-end industries in the surrounding region to cluster in the region. At the same time, due to the similar industrial structure, the neighboring regions have a certain degree of competition in economic development. With the aggravation of the aging problem in the region, the local industrial structure is forced to upgrade and adjust [22]. Due to the agglomeration effect of industries, the upgrading of the industrial structure in the region will often have a siphoning effect on the high-end industries in the surrounding areas [23]. Thus, the space for adjustment and upgrading of industrial structure in neighboring regions is limited, and the cultivation and development of high-end industries related to low-carbon emissions in neighboring regions are affected. In addition, the deepening of the aging population has increased the environmental requirements of the region. In the process of transformation and upgrading of regional industrial structure, some low-end industries with high energy consumption and high emission are usually transferred to neighboring regions [24]. Under the role of industrial agglomeration and sharing mechanism, this transfer may lead to the further agglomeration of low-end industries in neighboring regions, so that the high-emission low-end enterprises that should be upgraded and eliminated will continue to stay in the region. This situation has a negative impact on the overall industrial upgrading and emission reduction measures in neighboring regions, and further increases the carbon emission of neighboring regions [25]. Based on the above analysis, this paper proposes hypothesis 2:

H2: Population aging will increase the carbon emission of neighboring areas

## 2.2 Indirect effects of ageing on carbon emission

**2.2.1 The mediating effect of household consumption.**   Compared with other age groups, the elderly population is relatively less active in social consumption due to the decline in physical function [12]. Existing studies have verified this conclusion by introducing the consumption function of PA-age dependency ratio and analyzing regional data. At the same time, in China, the elderly generally have the motivation to bequeathing wealth to their offspring [26]. Driven by this motivation, the elderly tend to reduce their own consumption, and the property bequeathing to their offspring is often not easy to realize quickly. Thus, it has a negative impact on the overall consumption of society. In addition, aging will lead to the increase of non-productive expenditure in national income and squeeze the expenditure of productive investment, which will have a negative impact on total output and per capita income, and eventually lead to the decline of per capita consumption growth rate [27]. To sum up, aging has a certain inhibitory effect on residents' consumption. Based on previous studies, residents' consumption level is positively correlated with carbon emission [28]. Therefore, the decrease in residents' consumption brought about by population aging may be an important path to influence carbon emission [29]. Based on the above analysis, hypothesis 3 is proposed in this paper:

H3: Population aging reduces carbon emission by reducing household consumption

**2.2.2 The mediating effect of new urbanization.**   At present, China's urbanization rate has exceeded 65%, and urbanization has entered a new stage. According to previous studies, the aging of urban and rural areas has different impacts on urbanization. The aging of rural population will inhibit urbanization through the pressure of family pension and the reduction of the supply of working-age labor, while the aging of urban population will promote the silver economy and bring more jobs by increasing the demand for related pension products from

the demand side. Thus, the population is attracted to further gather in cities and towns [30]. At present, with the development of urbanization in China entering a new stage, the proportion of the elderly population in urban areas is increasing year by year, and its promoting effect on urbanization is also becoming increasingly apparent, so that the intensification of aging on the whole begins to show its promoting effect on urbanization [31]. The population agglomeration brought about by the improvement of urbanization is conducive to the implementation of shared travel mode and the centralized utilization of energy, improving the overall energy use efficiency and further optimizing the industrial structure [32]. At the same time, population agglomeration brought by urbanization will have a positive impact on knowledge sharing, technological progress, economies of scale and other aspects, and will also help promote the upgrading of regional industrial structure and the improvement of relevant environmental protection systems [33]. Thus, the overall regional carbon emission efficiency can be improved and carbon emission can be reduced [34]. Based on the above analysis, this paper proposes hypothesis 4:

H4: Population aging reduces carbon emission by promoting new urbanization

**2.2.3 The mediating effect of environmental regulation.** At different stages of the life cycle, individuals' income levels, lifestyles and attitudes towards environmental issues are different [35]. Due to the enhanced awareness of health risks, deeper cognition of environmental quality, and changes in lifestyle and needs, the elderly group has lower tolerance and higher requirements for environmental quality in the face of environmental pollution than the young group [36]. Subjectively, they are also more willing to push the government to develop more environmental protection measures through actions. Therefore, population aging has a positive impact on improving the intensity of regional environmental regulation. According to the Porter hypothesis, the improvement of environmental regulation intensity will encourage enterprises to carry out more innovative research and development activities, drive the reform of corresponding production technology and upgrade of environmental protection technology [37]. Thus, the energy saving and carbon reduction level of enterprises can be improved [38]. In addition, the enhancement of environmental regulation intensity will also help to further enhance the public's awareness of environmental protection, guide the society to form a low-carbon and environmentally friendly life concept, and guide the public to spontaneously strengthen supervision over high-emission enterprises [39], thus promoting the development of low-carbon emission reduction in the whole society. Based on the above analysis, this paper proposes hypothesis 5:

H5: Population aging can reduce carbon emission by improving environmental regulations.

## 3. Materials and methods

### 3.1 Variable description

**3.1.1 Dependent variable.** The dependent variable of this paper is carbon emission($CO2$). Carbon dioxide emissions are calculated using the consumption of eight energy sources including coal, coke, crude oil, gasoline, kerosene, diesel, fuel oil, and natural gas according to the IPCC carbon emissions calculation method (1) [40, 41].

$$Co_i = \sum_{i=1}^{8} (Co_2)_i = \sum_{i=1}^{8} E_i \times SCC_i \times CEF_i \tag{1}$$

Where, $Co_i$ is the estimated value of carbon emission; i is the type of fossil energy, i = 1, 2,..., 8; $E_i$ is the consumption of each fossil energy; $SCC_i$ is the discount coal coefficient of fossil energy. $CEF_i$ is the carbon emission factor.

The carbon intensity of China's provinces is then calculated using the ratio of $CO_2$ emissions to regional GDP. The formula is

$$Ci_j = \frac{Co_j}{GDP_j}, j = 1, 2, \ldots\ldots, 30 \tag{2}$$

Where, j stands for observation province, CI stands for carbon emission; Co stands for carbon dioxide emissions; GDP represents the gross regional product, and the data is from the "China Statistical Yearbook."

**3.1.2 Explanatory variables.** The explanatory variable of this paper is population aging (PA), and the proportion of the population aged 65 and above is selected as the index of population aging (PA), and the elderly dependency ratio is used to test the robustness of the test.

**3.1.3 Mediation variables.**

1. Residential consumption (RC): Existing studies on consumption often use the total retail sales of consumer goods as a variable to measure consumption. However, the disadvantage of this method is that the total retail sales of consumer goods include government purchases. Therefore, from the perspective of measuring residents' consumption, this paper uses the per capita consumption expenditure of each province (ten thousand yuan) as a metric.

2. New urbanization rate (UR): The core of new urbanization lies in population. Based on relevant studies, this paper uses the ratio of urban population to permanent resident population to represent the new urbanization rate of this region.

3. Environmental regulation (ER): Different scholars measure environmental regulation (ER) according to different indicators, including the number of penalties in environmental protection cases or the number of proposals related to the environment from a micro perspective, and the proportion of industrial pollution control investment in the secondary industry from a macro perspective, and the environmental tax related to pollution discharge fees and market incentives. This research starts from the perspective of pollution control results and considers the overall impact on the overall environment and carbon emissions. Finally, entropy method is adopted to build a comprehensive index of environmental regulation intensity by calculating the data of industrial wastewater, industrial SO2 and industrial soot emissions. Thus, the intensity of environmental regulation can be measured more comprehensively and objectively.

**3.1.4 Control variables.** This study controls a series of variables affecting carbon emission, including the level of economic development, which has a great impact on green carbon emission. A large number of scholars have found that the intensity of carbon emissions is often closely related to the level of local economic development. In this paper, per capita gross regional product (PGDP) is selected as an indicator to measure the level of local economic development. Degree of openness. With the increasing pressure of energy and environment in China, the external environment also has an important impact on China's pollution emission. This paper uses the total investment of foreign-invested enterprises (OPEN) to measure the degree of opening to the outside world. Science and education level. Regions endowed with a robust science and education level tend to experience a relatively faster and higher-quality

economic development, which subsequently impacts regional carbon emission. As a proxy for the level of science and education within a region, this paper employs the average number of students enrolled in higher education institutions per 100,000 population (EDU) as an indicator to quantify the region's science and education attainments. Government intervention: Local government's intervention in development will affect economic activities related to carbon emissions. Here, the ratio of local general budget expenditure to regional GDP is selected as the measure of government intervention (GI). Urban construction, all kinds of economic activities in the process of urban construction are closely related to carbon emissions. This paper selects the urban construction land area (UC) as a measurement index. Economic structure, the difference of regional economic structure also plays an important role in carbon emissions. The ratio of total retail sales of consumer goods to gross regional product (ES) is selected to measure economic structure.

## 3.2 Data sources

The data used in this paper were from "China Statistical Yearbook", "China Energy Statistical Yearbook" and "China Environmental Statistical Yearbook". Specifically, data related to carbon emission come from the "China Energy Statistics Yearbook", data related to environmental regulation on three-waste emissions come from the "China Environment Statistics Yearbook", and all other variables come from the" China Statistics Yearbook "of previous years. Considering the availability and completeness of data, Hong Kong, Macao, Taiwan and Tibet Autonomous Region of China were deleted, and the interpolation method was used to process some missing data. Finally, we select the panel data of 30 provinces in China from 2004 to 2022 as the basis of the study. The descriptive statistical results of each variable are shown in Table 1.

## 3.3 Spatial-temporal characteristics of carbon emission and aging

Based on the index system constructed in this paper, to enhance the intuitiveness of the results, we selected three time points: 2004, 2012, and 2022. The breakpoint method was employed to classify both carbon emission and the degree of aging into five distinct levels. Subsequently, ArcGIS software was utilized to generate distribution maps illustrating the carbon emission and aging degrees.

As depicted in Fig 2, during the sample period, China's carbon emission exhibited an overall steady decline. Notably, the carbon emission displayed geographical disparities, with higher

**Table 1. Descriptive statistics.**

| Variable | N | Mean | Sd | Median | Min | Max |
|---|---|---|---|---|---|---|
| CO2 | 570.00 | 2.86 | 2.22 | 2.18 | 0.17 | 14.40 |
| PA | 570.00 | 0.10 | 0.02 | 0.10 | 0.05 | 0.19 |
| RC | 570.00 | 1.40 | 0.84 | 1.27 | 0.27 | 4.89 |
| UR | 570.00 | 0.56 | 0.14 | 0.55 | 0.25 | 0.90 |
| ER | 510.00 | 0.51 | 0.52 | 0.32 | 0.00 | 2.59 |
| lnPGDP | 570.00 | 10.48 | 0.72 | 10.57 | 8.35 | 12.16 |
| lnOPEN | 570.00 | 11.01 | 1.58 | 10.97 | 6.55 | 15.49 |
| lnEDU | 570.00 | 7.78 | 0.38 | 7.77 | 6.58 | 8.92 |
| GI | 570.00 | 0.23 | 0.11 | 0.21 | 0.09 | 0.76 |
| lnUC | 570.00 | 7.10 | 0.78 | 7.19 | 4.65 | 8.67 |
| ES | 570.00 | 0.38 | 0.07 | 0.38 | 0.18 | 0.61 |

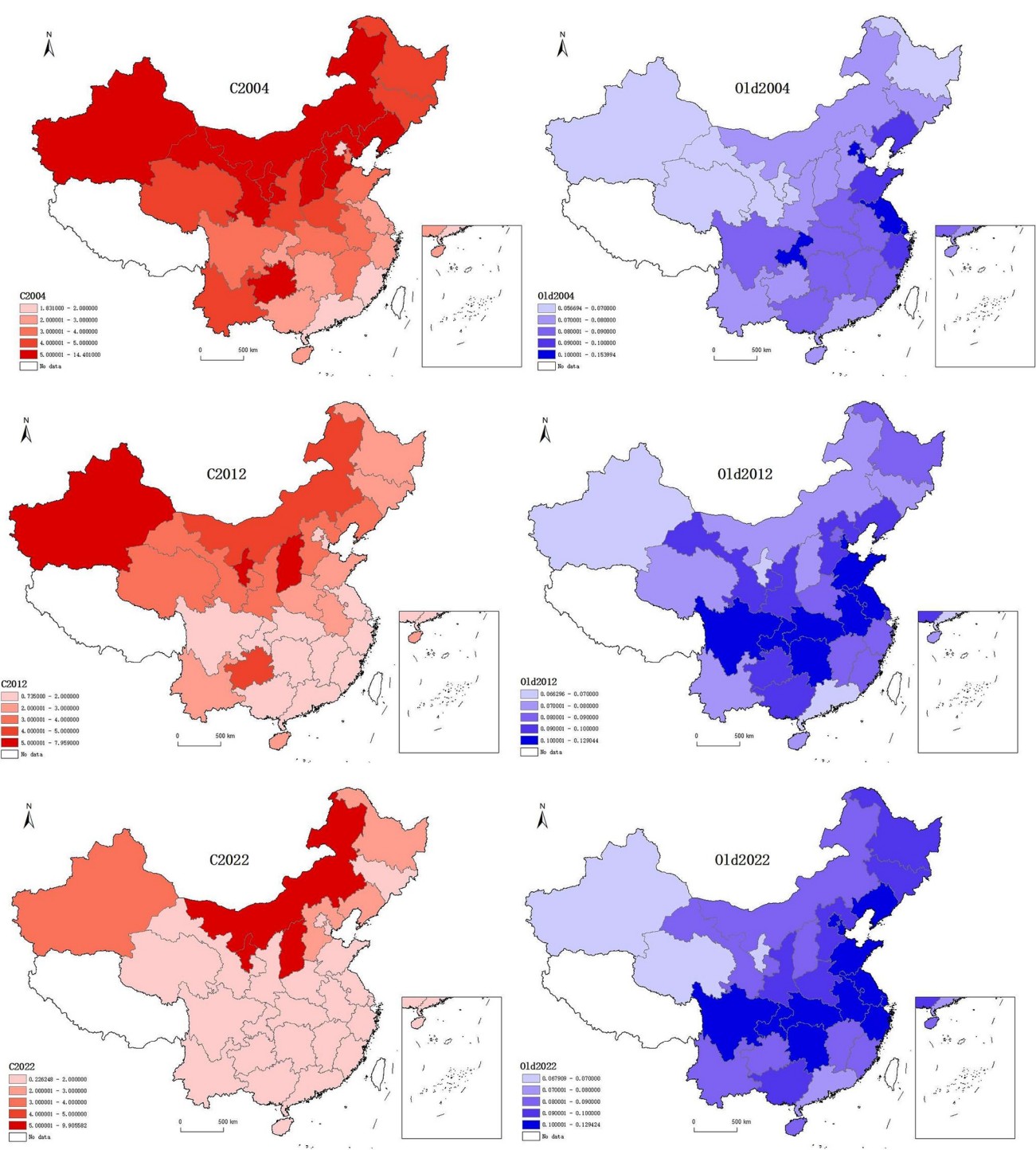

**Fig 2. Spatial Distribution of carbon emission and aging in China, 2004, 2012 and 2022.**

intensities prevalent in resource-rich provinces in the western and northeastern regions, and comparatively lower intensities observed in the eastern coastal provinces. Additionally, nation-wide, carbon emission significantly decreased over time, except for resource-based provinces in the central and western parts of the country, including Inner Mongolia and Shanxi. By

2022, the carbon emission of major provinces in southern China had almost entirely shifted to the lowest range. China's population aging is gradually deepening, with higher levels of aging geographically concentrated in some of the eastern coastal and southern provinces. Except for a few western provinces such as Xinjiang and Qinghai, there is a clear trend of deepening aging throughout the country. In 2004, the deeply aging provinces included only a few provinces such as Jiangsu, Beijing, Tianjin, and Chongqing, but by 2012, the range had expanded significantly, with many eastern and central-western provinces such as Shandong, Anhui, Sichuan, and Hunan entering the deeply aging range one after another. By 2022, the range has further expanded, and the level of aging has risen significantly across the Northeast region, with Liaoning Province also entering the deep aging zone.

## 4. Spatial correlation test and model selection

### 4.1 Global spatial correlation test

The Moran 'i test for spatial correlation of carbon emission of various provinces in China during 2004–2022 was conducted. The specific calculation formula is as follows:

$$Moran'I = \frac{n\sum_{i=1}^{n}\sum_{j=1}^{n}W(x_i - \bar{x})(x_j - \bar{x})}{\sum_{i=1}^{n}\sum_{j=1}^{n}W\sum_{i=1}^{n}(x_i - \bar{x})^2} \tag{3}$$

In this formula, n represents 30 provinces in China, $x_i$ and $\bar{x}$ respectively represent the carbon emission of each province and its mean value. Moran's I∈[−1,1], the greater the absolute value of Moran index, the stronger the spatial correlation of carbon emission. W is a spatial weight matrix, whose main function is to indicate the spatial connections among research objects. The premise of spatial autocorrelation test is to set a reasonable spatial weight matrix [42].

In this paper, geographical adjacency matrix, geographical distance matrix and economic distance matrix are used as three kinds of spatial weight matrices. The geographic adjacency matrix is divided based on whether two cities are directly adjacent to each other. If two cities have a common border, the value in the geographic adjacency matrix is 1, otherwise it is 0. Considering that there may be deviation in measuring the spatial correlation between regions only by whether the space is adjacent, the geographical distance matrix and the economic distance matrix are constructed. Geographical distance matrix is the inverse of the distance between two regions to form a value in the spatial weight matrix, generally using the distance between the capital cities of two provinces. The economic distance matrix reflects the economic correlation between different provinces and is constructed according to the GNP per capita between them.

Table 2 shows the global Moran's I value of carbon emission under three different weight matrices, and it can be found that the carbon emission of each province has obvious spatial correlation.

### 4.2 Local spatial correlation test

In this part, the local Moran's index scatter plot is output to show the agglomeration type of carbon emission among provinces. Fig 3 shows the Moran's I scatter plot of carbon emission for 2004 and 2022 based on the geographical distance and economic distance matrix. Under the two weight matrices, the distribution of carbon emission is mainly concentrated in the first and third quadrants, that is, the state of low and low concentration (LL) and high

**Table 2. Global Moran's I index of carbon emission of 30 provinces from 2004 to 2022.**

| Year | Spatial adjacency matrix | | Geographic distance matrix | | Economic distance adjacency matrix | |
|---|---|---|---|---|---|---|
| | Moran'I | p-value | Moran'I | p-value | Moran'I | p-value |
| 2004 | 0.266 | 0.002 | 0.061 | 0.003 | 0.129 | 0.049 |
| 2005 | 0.278 | 0.002 | 0.067 | 0.002 | 0.153 | 0.030 |
| 2006 | 0.267 | 0.002 | 0.066 | 0.002 | 0.169 | 0.021 |
| 2007 | 0.284 | 0.001 | 0.069 | 0.002 | 0.197 | 0.011 |
| 2008 | 0.351 | 0.000 | 0.090 | 0.000 | 0.216 | 0.007 |
| 2009 | 0.323 | 0.000 | 0.085 | 0.000 | 0.212 | 0.008 |
| 2010 | 0.348 | 0.000 | 0.087 | 0.000 | 0.203 | 0.010 |
| 2011 | 0.337 | 0.000 | 0.083 | 0.000 | 0.175 | 0.018 |
| 2012 | 0.348 | 0.000 | 0.090 | 0.000 | 0.179 | 0.016 |
| 2013 | 0.333 | 0.000 | 0.089 | 0.000 | 0.163 | 0.023 |
| 2014 | 0.337 | 0.000 | 0.086 | 0.000 | 0.147 | 0.034 |
| 2015 | 0.312 | 0.000 | 0.079 | 0.000 | 0.120 | 0.059 |
| 2016 | 0.327 | 0.000 | 0.081 | 0.000 | 0.125 | 0.056 |
| 2017 | 0.392 | 0.000 | 0.061 | 0.003 | 0.021 | 0.288 |
| 2018 | 0.366 | 0.000 | 0.083 | 0.000 | 0.092 | 0.099 |
| 2019 | 0.404 | 0.000 | 0.090 | 0.000 | 0.095 | 0.095 |
| 2020 | 0.408 | 0.000 | 0.085 | 0.000 | 0.087 | 0.110 |
| 2021 | 0.314 | 0.000 | 0.068 | 0.001 | 0.054 | 0.098 |
| 2022 | 0.301 | 0.000 | 0.064 | 0.001 | 0.052 | 0.173 |

concentration (HH) is basically significant, indicating that there is a homogeneity of carbon emission distribution in local space.

## 4.3 Model setting

**4.3.1 STIRPAT model.** STIRPAT model has been widely used in the analysis of the impact of economic and social factors on environmental quality [43], and this model has good scalability. Currently, some scholars have included aging in this model to analyze the impact of aging on carbon emission. Based on the above analysis, this paper constructs the STIRPAT theoretical analysis model of Eq (4).

$$Ci_{it} = \lambda_0 + \lambda_1 PA_{it} + \lambda_3 Control_{it} + \mu_i + \eta_t + \varepsilon_{it} \tag{4}$$

Where i = 1,2,3,..., 30, representing 30 provinces in China, t = 1,2,3,..., 17, represents the time variable from 2004 to 2022, $Ci_{it}$ and $PA_{it}$ respectively represent the carbon emission and aging degree of t years in Province i. $Control_{it}$ is a set of control variables including GDP per capita, where $\mu_i$ represents city effect; $\eta_t$ represents the time effect; $\varepsilon_{it}$ represents the random error term.

**4.3.2 Spatial econometric model selection.** In terms of model selection, it referred to the research of [44], conducted LM, LR, Hausman test and Wald test in turn, and the results passed the significance test, so as to determine the spatial Dubin model with double fixed effects. Further combined with the research of Tang Kaitao, Shao Shuai et al., the spatial Durbin model is set as follows:

$$Ci_{it} = \lambda_0 + \rho_1 \sum_j w_{ij} Ci_{it} + \lambda_1 PA_{it} + \lambda_2 \sum_j w_{ij} old_{it}$$
$$+ \lambda_3 Control_{it} + \lambda_4 \sum_j w_{ij} Control_{it} + \mu_i + \eta_t + \varepsilon_{it} \tag{5}$$

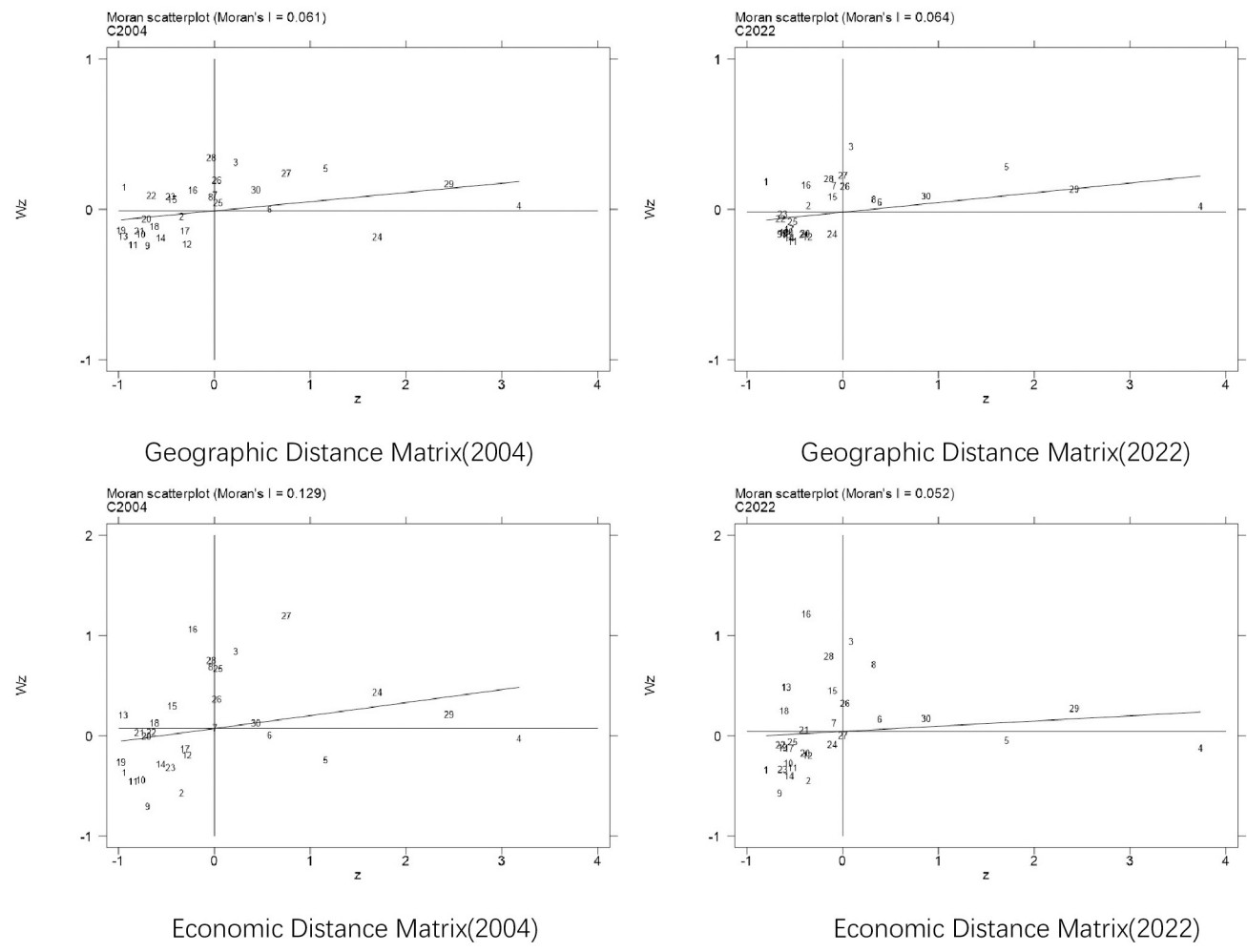

**Fig 3. Local Moran's I scatter plots of carbon emission in 2004 and 2022.**

Where: i and j represent cities; t is the year; $C_{it}$ represents carbon emission, and $\rho$ to be estimated measures the spatial spillover effect of regional carbon emission. $PA_{it}$ represents the level of aging, the estimated coefficient $\lambda_1$ measures the effect of aging on carbon emission in the province, and the estimated coefficient $\lambda_2$ represents the spatial spillover effect of aging. wij represents the spatial weight matrix, including geographical adjacency space weight matrix, geographical distance space weight matrix and economic distance space weight matrix and economy. Control is a set of control variables; $\mu_i$ represents regional effect; $\eta_t$ represents the time effect; $\varepsilon_{it}$ represents the random error term.

**4.3.3. Mediator effect test model.** Further, spatial mediating effect was used to explore whether household consumption, new-type urbanization and environmental regulation had mediating effect on the influence of aging on carbon emission. The specific model setting form is as follows:

$$
\begin{aligned}
Medi_{it} &= \alpha_0 + \rho_2\sum_j w_{ij}Med_{it} + \alpha_1 PA_{it} + \alpha_2\sum_j w_{ij}PA_{it} \\
&+ \alpha_3 Control_{it} + \alpha_4\sum_j w_{ij}Control_{it} + \mu_i + \eta_t + \varepsilon_{it}
\end{aligned} \tag{6}
$$

$$Ci_{it} = \beta_0 + \rho_3 \sum_j w_{ij} Ci_{it} + \beta_1 PA_{it} + \beta_2 \sum_j w_{ij} PA_{it} + \beta_3 Med_{it}$$
$$+ \beta_4 \sum_j w_{ij} Med_{it} + \beta_5 Control_{it} + \beta_6 \sum_j w_{ij} Control_{it} + \mu_i + \eta_t + \varepsilon_{it} \tag{7}$$

In formula (6), $Med_{it}$ represents channel variable, including three variables: residential consumption (RC), new urbanization (UR) and environmental regulation (ER). The influence of aging on the intermediary variable is verified according to formula (6). Then, according to formula (7), we can determine whether the intermediary effect exists.

## 5. Results and discussion

### 5.1 Spatial datum regression

Table 3 shows the estimation results of the spatial Durbin model. Under the three different spatial weight matrices, the estimated coefficients of aging on carbon emission are all significantly negative, indicating that aging is conducive to promoting regional emission reduction. The estimated coefficient of the spatial lag variable (W·PA) of aging is significantly positive, which indicates that aging will have an impact on carbon emissions in the surrounding areas through the spatial spillover effect.

For the spatial econometric model, the total effect is decomposed into direct effect and indirect effect, and the decomposition results are shown in Table 4. As can be seen from Table 4, the direct effect of population aging is negative and significant at 1% level, indicating that it can significantly reduce the carbon emission of the province. On the one hand, with the growth of age, people's participation in social and economic activities will be reduced to a certain extent. At the same time, the elderly people are affected by the thought of hardship and simplicity in the past, and their living habits and personal consumption are relatively economical compared with the young people, thus relatively reducing the behavior of high carbon emissions. On the other hand, with the decline in the number of working people and the development of social science and education level brought by the aging population, the driving force of economic development has also changed from the "first demographic dividend" relying solely on the number of labor forces to the "second demographic dividend" relying on the quality of labor forces.

When the aging of the population leads to the reduction of the labor force, many provinces begin to promote further economic development through the transformation of economic momentum, and gradually shift from the previous high-emission and high-pollution industries to high-tech industries, which also objectively reduces the carbon emission of the region. The transformation of the province's economic development may attract more high-tech industries from the surrounding areas to the province, and at the same time make some high-emission and high-pollution industries transfer to neighboring provinces, thus increasing the carbon emission of neighboring provinces. Therefore, the indirect effect of population aging is positive, that is, population aging will increase the carbon emission of the surrounding areas.

In addition to the degree of ageing, we also consider other factors that may potentially influence the carbon emission within the province as well as in neighboring provinces. The level of local economic development (lnGDP) has no significant impact on the province's carbon emission, yet it significantly reduces the carbon emission of neighboring provinces. This phenomenon can be attributed to the effects of technology transfer and industrial cooperation. The degree of openness (lnOPEN) and urban construction (UC) do not significantly influence carbon emission. The enhancement of science and education (lnEDU) has a pronounced negative effect on the province's carbon emission, suggesting that technological innovation and educational advancement contribute to reducing carbon emissions within the region.

**Table 3. Estimation results of spatial Durbin model.**

| Variable | Geographical adjacency | Geographical distance | Economic distance |
|---|---|---|---|
| Main | | | |
| PA | -6.981** | -7.344** | -7.220** |
| | (3.103) | (3.137) | (3.017) |
| lnPGDP | -0.400 | -0.402 | -0.489 |
| | (0.314) | (0.366) | (0.368) |
| lnOPEN | 0.078 | 0.053 | 0.032 |
| | (0.071) | (0.073) | (0.070) |
| lnEDU | -1.373*** | -1.433*** | -1.360*** |
| | (0.292) | (0.295) | (0.309) |
| GI | -0.950 | -0.938 | -1.343 |
| | (0.880) | (0.894) | (0.946) |
| lnUC | -0.012 | -0.068 | 0.034 |
| | (0.273) | (0.266) | (0.267) |
| ES | -2.741*** | -2.414*** | -2.311*** |
| | (0.619) | (0.649) | (0.703) |
| W· | | | |
| W·PA | 9.093*** | 8.157* | 9.304** |
| | (3.528) | (4.653) | (3.744) |
| W·lnPGDP | -1.261*** | -1.788*** | -1.319*** |
| | (0.393) | (0.640) | (0.475) |
| W·lnOPEN | 0.331** | 0.718** | 0.324** |
| | (0.149) | (0.319) | (0.159) |
| W·lnEDU | 0.631 | 2.072** | 0.239 |
| | (0.504) | (0.906) | (0.600) |
| W·GI | -3.345** | -2.937 | -2.620 |
| | (1.563) | (2.764) | (1.648) |
| W·lnUC | -0.524 | -3.108** | 0.267 |
| | (0.523) | (1.362) | (0.532) |
| W·ES | 0.023 | 4.208 | -1.611 |
| | (0.838) | (2.624) | (1.365) |
| Spatial | | | |
| rho | -0.143** | -0.476*** | -0.188*** |
| | (0.058) | (0.161) | (0.071) |
| Variance | | | |
| sigma2_e | 0.393*** | 0.390*** | 0.400*** |
| | (0.023) | (0.023) | (0.024) |
| N | 570 | 570 | 570 |
| R2 | 0.326 | 0.310 | 0.254 |

However, it shows a positive effect in neighboring regions, which may mirror the competition for resources and the environmental pressures among these regions. Government intervention (GI) has a non-significant impact on the province's carbon emission but exerts a significant negative effect on the carbon emission of neighboring provinces, highlighting the government's positive role in fostering regional environmental cooperation and mitigating carbon emissions. Economic structure (ES) primarily exerts a negative influence on the province's carbon emission, while their effect on neighboring regions is insignificant. This could be linked to

**Table 4. Decomposition results of spatial Durbin model.**

| Variable | Geographical adjacency | Geographical distance | Economic distance |
|---|---|---|---|
| Main | | | |
| PA | -6.981** | -7.344** | -7.220** |
| | (3.103) | (3.137) | (3.017) |
| lnPGDP | -0.400 | -0.402 | -0.489 |
| | (0.314) | (0.366) | (0.368) |
| lnOPEN | 0.078 | 0.053 | 0.032 |
| | (0.071) | (0.073) | (0.070) |
| lnEDU | -1.373*** | -1.433*** | -1.360*** |
| | (0.292) | (0.295) | (0.309) |
| GI | -0.950 | -0.938 | -1.343 |
| | (0.880) | (0.894) | (0.946) |
| lnUC | -0.012 | -0.068 | 0.034 |
| | (0.273) | (0.266) | (0.267) |
| ES | -2.741*** | -2.414*** | -2.311*** |
| | (0.619) | (0.649) | (0.703) |
| W· | | | |
| W·PA | 9.093*** | 8.157* | 9.304** |
| | (3.528) | (4.653) | (3.744) |
| W·lnPGDP | -1.261*** | -1.788*** | -1.319*** |
| | (0.393) | (0.640) | (0.475) |
| W·lnOPEN | 0.331** | 0.718** | 0.324** |
| | (0.149) | (0.319) | (0.159) |
| W·lnEDU | 0.631 | 2.072** | 0.239 |
| | (0.504) | (0.906) | (0.600) |
| W·GI | -3.345** | -2.937 | -2.620 |
| | (1.563) | (2.764) | (1.648) |
| W·lnUC | -0.524 | -3.108** | 0.267 |
| | (0.523) | (1.362) | (0.532) |
| W·ES | 0.023 | 4.208 | -1.611 |
| | (0.838) | (2.624) | (1.365) |
| Spatial | | | |
| rho | -0.143** | -0.476*** | -0.188*** |
| | (0.058) | (0.161) | (0.071) |
| Variance | | | |
| sigma2_e | 0.393*** | 0.390*** | 0.400*** |
| | (0.023) | (0.023) | (0.024) |
| N | 570 | 570 | 570 |
| R2 | 0.326 | 0.310 | 0.254 |
| Variable | Geographical adjacency | Geographical distance | Economic distance |
| Direct effects | | | |
| PA | -7.191** | -7.463** | -7.487** |
| | (3.281) | (3.318) | (3.219) |
| lnPGDP | -0.374 | -0.387 | -0.463 |
| | (0.311) | (0.363) | (0.370) |
| lnOPEN | 0.075 | 0.046 | 0.028 |
| | (0.069) | (0.070) | (0.069) |
| lnEDU | -1.407*** | -1.491*** | -1.380*** |
| | (0.284) | (0.289) | (0.301) |

*(Continued)*

**Table 4.** (Continued）

| | | | |
|---|---|---|---|
| GI | -0.833 | -0.885 | -1.256 |
| | (0.876) | (0.892) | (0.942) |
| lnUC | 0.026 | 0.014 | 0.044 |
| | (0.275) | (0.265) | (0.272) |
| ES | -2.760*** | -2.529*** | -2.282*** |
| | (0.650) | (0.686) | (0.771) |
| Indirect effects | | | |
| PA | 8.944** | 7.904** | 9.136** |
| | (3.506) | (4.018) | (3.674) |
| lnPGDP | -1.065*** | -1.078** | -1.047** |
| | (0.382) | (0.511) | (0.452) |
| lnOPEN | 0.287** | 0.486** | 0.275** |
| | (0.132) | (0.207) | (0.139) |
| lnEDU | 0.724 | 1.880*** | 0.405 |
| | (0.461) | (0.652) | (0.522) |
| GI | -2.895** | -1.646 | -2.045 |
| | (1.430) | (2.039) | (1.499) |
| lnUC | -0.519 | -2.221** | 0.192 |
| | (0.463) | (0.913) | (0.459) |
| ES | 0.409 | 3.714** | -1.017 |
| | (0.771) | (1.847) | (1.274) |

the fact that a consumption-driven economy diminishes the province's reliance on heavy industry and energy-intensive sectors.

## 5.2 Robustness test

This paper empirically tests the impact of population aging on carbon emission by using the dual control spatial Durbin model. It is found that population aging is not only conducive to reducing carbon emission in the region, but also has a significant spatial effect, affecting the carbon emission level in neighboring regions. In order to test the robustness of the above estimation results, this part will conduct robustness tests by replacing explanatory variables and narrow-sample processing respectively

**5.2.1 Replace explanatory variables.**   In order to better test the results of spatial regression, robustness was verified by substituting explanatory variables for aging. The aging variable was replaced by the proportion of the population aged 65 and above with the old-age dependency ratio, and the three spatial weight matrices were used again to perform spatial regression on the samples. The regression results are shown in the first three columns of Table 5. Under the three spatial matrices, the influence of aging on the estimated coefficient of carbon emissions is still significantly negative, which passes the robustness test.

**5.2.2 Narrow sample processing.**   In order to avoid the influence of sample extreme values on the regression results, the lowest 1% and the highest 1% are excluded in this part for aging. The estimated results are shown in the last three columns of Table 5. The results of aging on carbon emissions are still significantly negative, and the spatial lag variable coefficient of aging is significantly positive, which is consistent with the regression results and has passed the robustness test.

**Table 5. Robustness test.**

| Variable | Replace explanatory variables | | | Tail reduction | | |
|---|---|---|---|---|---|---|
| | Geographical adjacency | Geographical distance | Economic distance | Geographical adjacency | Geographical adjacency | Geographical adjacency |
| PA | -3.703* | -4.031* | -3.603* | -7.046** | -7.356** | -7.220** |
| | (2.169) | (2.169) | (2.144) | (3.133) | (3.176) | (3.062) |
| W·PA | 7.926** | 4.748 | 7.230** | 9.433*** | 8.206* | 9.316** |
| | (3.329) | (6.282) | (3.245) | (3.588) | (4.725) | (3.791) |
| Direct effects | -4.058* | -4.347* | -2.336 | -7.266** | -7.475** | -7.485** |
| | (2.303) | (2.285) | (2.009) | (3.312) | (3.358) | (3.267) |
| Indirect effects | 7.376** | 10.975** | 6.483** | 9.256*** | 7.939* | 9.144** |
| | (2.909) | (4.584) | (2.920) | (3.558) | (4.069) | (3.720) |
| Control variables | YES | YES | YES | YES | YES | YES |
| Area | YES | YES | YES | YES | YES | YES |
| Year | YES | YES | YES | YES | YES | YES |
| N | 570 | 570 | 570 | 570 | 570 | 570 |
| R2 | 0.296 | 0.280 | 0.260 | 0.326 | 0.308 | 0.253 |

## 5.3 Time effect analysis

Although static regression analysis shows that aging is conducive to reducing regional carbon emission, considering the dynamic aspects, how future aging changes will continue to have an impact on regional carbon emission and what kind of change trend this impact presents, further discussion needs to be carried out based on the actual situation. With the deepening trend of population aging today, with the improvement of people's income and living standards, more attention to national policies on carbon emissions and the launch of relevant low-carbon environmental protection policies, what dynamic changes will the impact of aging on carbon emission usher in over time?

$$Ci_{it} = \lambda_0 + \rho_1\sum_j w_{ij}Ci_{it} + \lambda_1 PA_{it} + \lambda_2\sum_j w_{ij}PA_t + \lambda_3 PA_{tt*}Year_{it}$$
$$+\lambda_4\sum_j w_{ij}PA_{tt*}Year_{it} + \lambda_5 Control_{it} + \lambda_6\sum_j w_{ij}Control_{it} + \mu_i + \eta_t + \varepsilon_{it} \quad (8)$$

In this paper, sample estimation is carried out from time dimension. Check using rolling time window analysis. The specific steps are as follows: ① Set the rolling time window to 10 years; ② Set a time dummy variable year, when the year is less than the midpoint of the time window, the year is 0 and the other is 1; On the basis of the original formula (5), the cross-multiplication term PA×year of aging and time dummy variables is added respectively, and the equation of formula (8) is further constructed. Then the rolling window estimation is carried out. The advantage of this is that the coefficient of aging can be more accurately reflected from a dynamic point of view.

Table 6 shows the change of the influence of aging on carbon emission under the geographical adjacency matrix under the rolling time window. It can be found that the coefficient of aging has been significantly negative in the whole time interval, indicating that the overall aging can still effectively reduce the regional carbon emission. However, in the whole time interval, the interaction term with time is positive, indicating that the emission reduction effect of aging on carbon emission is gradually declining with the passage of time. This phenomenon may be due to a combination of factors. On the one hand, economic and technological developments may make other factors, such as the adoption of cleaner energy sources and improvements in energy efficiency, more important drivers of carbon emission, thus reducing the

**Table 6. Rolling estimation results of the temporal effect of carbon emission under the geographic adjacency matrix.**

| Variable | 2004–2013 (2009 Midpoint) | 2005–2014 (2010 Midpoint) | 2006–2015 (2011 Midpoint) | 2007–2016 (2012 Midpoint) | 2008–2017 (2013 Midpoint) |
|---|---|---|---|---|---|
| | $CO_2$ | $CO_2$ | $CO_2$ | $CO_2$ | $CO_2$ |
| PA | -20.148*** | -18.824*** | -16.813*** | -14.869*** | -13.818*** |
| | (4.572) | (4.430) | (4.362) | (4.186) | (4.135) |
| PAyear1 | 13.826*** | | | | |
| | (3.571) | | | | |
| PAyear2 | | 12.668*** | | | |
| | | (3.401) | | | |
| PAyear3 | | | 10.267*** | | |
| | | | (3.232) | | |
| PAyear4 | | | | 8.199*** | |
| | | | | (3.031) | |
| PAyear5 | | | | | 7.158** |
| | | | | | (2.940) |
| Control variables | YES | YES | YES | YES | YES |
| Area | YES | YES | YES | YES | YES |
| Year | YES | YES | YES | YES | YES |
| N | 570 | 570 | 570 | 570 | 570 |
| R2 | 0.351 | 0.355 | 0.351 | 0.349 | 0.349 |
| | 2009–2018 (2014 Midpoint) | 2010–2019 (2015 Midpoint) | 2011–2020 (2016 Midpoint) | 2012–2021 (2017 Midpoint) | 2013–2022 (2018 Midpoint) |
| | $CO_2$ | $CO_2$ | $CO_2$ | $CO_2$ | $CO_2$ |
| PA | -13.038*** | -12.124*** | -11.766*** | -10.831*** | -8.891** |
| | (4.080) | (3.967) | (3.899) | (3.828) | (3.734) |
| PAyear6 | 6.378** | | | | |
| | (2.856) | | | | |
| PAyear7 | | 5.572** | | | |
| | | (2.776) | | | |
| PAyear8 | | | 5.337** | | |
| | | | (2.696) | | |
| PAyear9 | | | | 4.515* | |
| | | | | (2.673) | |
| PAyear10 | | | | | 2.305 |
| | | | | | (2.690) |
| Control variables | YES | YES | YES | YES | YES |
| Area | YES | YES | YES | YES | YES |
| Year | YES | YES | YES | YES | YES |
| N | 570 | 570 | 570 | 570 | 570 |
| R2 | 0.348 | 0.345 | 0.344 | 0.341 | 0.333 |

direct effect of ageing. On the other hand, the effect of ageing on reducing carbon emission may also be offset to some extent by higher incomes and higher consumption levels of the older population over time. In addition, changes in the policy environment, such as the introduction of energy-saving and emission-reduction policies and an increase in people's awareness of environmental protection, may offset to some extent the reducing effect of ageing on carbon emissions.

Table 7 shows the change of the influence of aging on carbon emission under the geographical distance matrix under the rolling time window. The coefficient of aging has been significantly negative throughout the time interval, and the negative influence coefficient of aging has a decreasing trend as time goes on. It shows that aging can still effectively reduce the intensity of regional carbon emissions under the geographical adjacency matrix. The above conclusions are basically consistent with the results under the geographic adjacency matrix in Table 6, and the robustness of the results is further tested. The interaction term between aging

**Table 7. Rolling estimation results of the temporal effect of carbon emission under the geographical distance matrix.**

| Variable | 2004–2013 (2009 Midpoint) CO2 | 2005–2014 (2010 Midpoint) CO2 | 2006–2015 (2011 Midpoint) CO2 | 2007–2016 (2012 Midpoint) CO2 | 2008–2017 (2013 Midpoint) CO2 |
|---|---|---|---|---|---|
| PA | -21.121*** | -19.115*** | -16.032*** | -13.809*** | -12.772*** |
| | (4.882) | (4.729) | (4.640) | (4.414) | (4.334) |
| PAyear1 | 14.463*** | | | | |
| | (3.956) | | | | |
| PAyear2 | | 12.401*** | | | |
| | | (3.750) | | | |
| PAyear3 | | | 9.022** | | |
| | | | (3.556) | | |
| PAyear4 | | | | 6.810** | |
| | | | | (3.316) | |
| PAyear5 | | | | | 5.738* |
| | | | | | (3.192) |
| Control variables | YES | YES | YES | YES | YES |
| Area | YES | YES | YES | YES | YES |
| Year | YES | YES | YES | YES | YES |
| N | 570 | 570 | 570 | 570 | 570 |
| R2 | 0.366 | 0.363 | 0.356 | 0.341 | 0.330 |
| | 2009–2018 (2014 Midpoint) CO2 | 2010–2019 (2015 Midpoint) CO2 | 2011–2020 (2016 Midpoint) CO2 | 2012–2021 (2017Midpoint) CO2 | 2013–2022 (2018 Midpoint) CO2 |
| PA | -12.064*** | -10.782*** | -11.370*** | -10.987*** | -8.124** |
| | (4.266) | (4.110) | (4.059) | (3.975) | (3.841) |
| PAyear6 | 5.030 | | | | |
| | (3.097) | | | | |
| PAyear7 | | 3.961 | | | |
| | | (2.986) | | | |
| PAyear8 | | | 4.613 | | |
| | | | (2.916) | | |
| PAyear9 | | | | 4.383 | |
| | | | | (2.881) | |
| PAyear10 | | | | | 0.993 |
| | | | | | (2.879) |
| Control variables | YES | YES | YES | YES | YES |
| Area | YES | YES | YES | YES | YES |
| Year | YES | YES | YES | YES | YES |
| N | 570 | 570 | 570 | 570 | 570 |
| R2 | 0.321 | 0.314 | 0.320 | 0.317 | 0.312 |

and time is positive, but its coefficient and significance gradually decline with the passage of time. Since 2014 as the midpoint, the interaction term between aging and time is no longer significant, indicating that the emission reduction effect of aging on carbon emission is gradually declining with the passage of time, and the mitigation effect of aging on carbon emission is also weakening with the passage of time. The difference may be due to the information difference between the geographic distance matrix and the geographic adjacency matrix during construction. The geographical distance matrix also takes into account the physical distance between regions. Compared with the geographical adjacency matrix, even if two regions are close to each other in space, if they are far apart, they will be regarded as uncorrelated. Therefore, under the geographical distance matrix, the influence of aging on carbon emission may be limited by geographical distance, and the regions farther away will have a relatively small impact, which may objectively lead to a slight difference between the two.

Table 8 shows how the economic distance matrix is used to analyze the impact of aging on carbon emission under the rolling time window. The economic distance matrix further considers the economic links and exchanges between geographical locations, so it can better reflect the degree of economic interaction between different regions, and better measure the influence of aging on economic activities and consumption behaviors. How to further influence carbon emissions in the time trend. Considering the dynamic influence of time change, population aging can still effectively promote regional carbon emission reduction work, and its promoting effect on carbon emission reduction shows a trend of decline over time except for a slight rebound in the last period. This conclusion is basically consistent with the results in Tables 6 and 7. The interaction term with time is still positive, and the coefficient shows a trend of first decreasing and then increasing over time, indicating that under the economic distance matrix, the emission reduction effect of aging on carbon emission decreases first and then increases with the passage of time, but the general trend is basically declining. In summary, after analyzing the dynamic changes of regional carbon emission affected by population aging using geographical adjacency matrix, geographical distance matrix and economic distance matrix, it is found that aging has a promoting effect on regional carbon emission reduction from a dynamic perspective, but the promoting effect is weakened with the passage of time.

## 6. Extended analysis

### 6.1 Mediation effect

Based on the above theoretical analysis, population aging may reduce regional carbon emission by reducing household consumption, improving environmental regulations, and promoting new-type urbanization. In order to further test the path of the influence of aging on carbon emission, residential consumption (RC), new urbanization (UR) and environmental regulation (er) are selected as intermediary variables, and the mechanism test of these three channels is conducted using the spatial adjacency matrix based on Eqs (6) and (7).

Column (1) in Table 9 takes residents' consumption as the intermediary variable, in which the influence coefficient of aging on residents' consumption is -1.239, indicating that aging does have a restraining effect on residents' consumption level. In column (2), the influence coefficient of househPA consumption on carbon emission is 0.603, and is significant at 1% level, indicating that the increase of household consumption will increase the regional carbon emission. Combined with the results in column (1) and (2), it is found that aging further reduces the intensity of carbon emissions by inhibiting household consumption, and hypothesis H3 is valid. In Table 9, column (3) takes the new urbanization rate as the intermediary variable, in which the influence coefficient of aging on new urbanization is 0.227, indicating that aging does have a promoting effect on new urbanization. The influence coefficient of new-type

**Table 8. Rolling estimation results of the temporal effect of carbon emission under the economic distance matrix.**

| Variable | 2004–2013 (2009 Midpoint) | 2005–2014 (2010 Midpoint) | 2006–2015 (2011 Midpoint) | 2007–2016 (2012 Midpoint) | 2008–2017 (2013 Midpoint) |
|---|---|---|---|---|---|
| | $CO_2$ | $CO_2$ | $CO_2$ | $CO_2$ | $CO_2$ |
| PA | -20.150*** | -18.978*** | -16.580*** | -13.603*** | -12.092*** |
| | (4.715) | (4.631) | (4.591) | (4.296) | (4.202) |
| PAyear1 | 14.090*** | | | | |
| | (3.994) | | | | |
| PAyear2 | | 12.964*** | | | |
| | | (3.915) | | | |
| PAyear3 | | | 10.058*** | | |
| | | | (3.770) | | |
| PAyear4 | | | | 7.021** | |
| | | | | (3.486) | |
| PAyear5 | | | | | 5.423 |
| | | | | | (3.362) |
| Control variables | YES | YES | YES | YES | YES |
| Area | YES | YES | YES | YES | YES |
| Year | YES | YES | YES | YES | YES |
| N | 570 | 570 | 570 | 570 | 570 |
| R2 | 0.293 | 0.292 | 0.284 | 0.284 | 0.284 |
| | 2009–2018 (2014 Midpoint) | 2010–2019 (2015 Midpoint) | 2011–2020 (2016 Midpoint) | 2012–2021 (2017 Midpoint) | 2013–2022 (2018 Midpoint) |
| | $CO_2$ | $CO_2$ | $CO_2$ | $CO_2$ | $CO_2$ |
| PA | -11.029*** | -9.792** | -10.449*** | -10.380*** | -7.104* |
| | (4.110) | (3.913) | (3.856) | (3.790) | (3.673) |
| PAyear6 | 4.290 | | | | |
| | (3.246) | | | | |
| PAyear7 | | 3.080 | | | |
| | | (3.126) | | | |
| PAyear8 | | | 3.948 | | |
| | | | (3.060) | | |
| PAyear9 | | | | 3.911 | |
| | | | | (3.020) | |
| PAyear10 | | | | | -0.911 |
| | | | | | (3.015) |
| Control variables | YES | YES | YES | YES | YES |
| Area | YES | YES | YES | YES | YES |
| Year | YES | YES | YES | YES | YES |
| N | 570 | 570 | 570 | 570 | 570 |
| R2 | 0.281 | 0.274 | 0.276 | 0.277 | 0.268 |

urbanization on carbon emission is -4.786, and is significant at 1% level, indicating that the promotion of new-type urbanization is conducive to reducing carbon emission. China's urbanization as a whole has entered a new stage of development. Along with population agglomeration, it will bring positive externalities such as knowledge spillover and economies of scale, which will also be conducive to industrial upgrading and centralized utilization of energy, thus promoting regional energy conservation and emission reduction and effectively

**Table 9.  Analysis of mediating effect.**

| variable | (1) | (2) | (3) | (4) | (5) | (6) |
|---|---|---|---|---|---|---|
| | RC | CO2 | UR | CO2 | ER | CO2 |
| Main | | | | | | |
| PA | -1.152* | -6.284** | 0.227*** | -7.794** | 2.012* | -8.712** |
| | (0.657) | (3.070) | (0.074) | (3.086) | (1.125) | (3.553) |
| RC | | 0.603*** | | | | |
| | | (0.198) | | | | |
| UR | | | | -4.786*** | | |
| | | | | (1.750) | | |
| ER | | | | | | -0.261* |
| | | | | | | (0.138) |
| Control variables | YES | YES | YES | YES | YES | YES |
| Area | YES | YES | YES | YES | YES | YES |
| Year | YES | YES | YES | YES | YES | YES |
| N | 570 | 570 | 570 | 570 | 570 | 570 |
| R2 | 0.328 | 0.254 | 0.572 | 0.338 | 0.055 | 0.328 |

reducing carbon emission [45]. The empirical results in column (3) and (4) show that aging can further reduce regional carbon emission by promoting the development of new urbanization. Hypothesis H4 is valid. In Table 9, column (5) takes environmental regulation as the intermediary variable, in which the influence coefficient of aging on environmental regulation is 2.012, indicating that aging is conducive to strengthening regional environmental regulation. The influence coefficient of environmental regulation on carbon emission is -0.261, indicating that the improvement of environmental regulation intensity is conducive to the reduction of carbon emission. Due to their physical conditions and other objective reasons, the elderly have higher requirements than other groups in terms of environmental quality. Therefore, the increase in the number of elderly people brought about by aging will promote the increase of environmental regulation intensity. On the one hand, the improvement of environmental regulation intensity is conducive to the industrial upgrading and transformation of relevant high-emission enterprises [46]; on the other hand, it will further promote the improvement of the environmental awareness of the whole society, thus bringing a positive impact on the reduction of regional carbon emission [47]. Aging can further reduce regional carbon emission by improving the level of environmental regulation. Hypothesis H5 is valid.

## 6.2 Heterogeneity analysis

In order to further explain the regional differentiation of population aging on carbon emission, this paper further divides 30 provinces into three regions: eastern, central and western regions for heterogeneity analysis. According to the viewpoint of traditional economic geography, economically developed areas and economically underdeveloped areas are often faced with a large difference in location advantage. China's central and western inland cities have obvious differences in the degree of aging and carbon emission compared with the eastern coastal cities. Therefore, location factors should be considered when analyzing the influence of aging on carbon emission. According to the geographical location of the 30 provinces, the eastern region includes Beijing, Tianjin, Hebei, Liaoning, Shanghai, Jiangsu and other 12 provinces; The central region includes nine provinces, including Shanxi, Inner Mongolia, Jilin, Heilongjiang and

**Table 10. Heterogeneity in the eastern region.**

| variable | (1) | (2) | (3) | (4) | (5) | (6) | (7) |
|---|---|---|---|---|---|---|---|
| | CO2 | RC | CO2 | UR | CO2 | ER | CO2 |
| Main | | | | | | | |
| PA | -2.636 | -1.745** | -0.423 | 0.525*** | 2.494 | 6.359*** | -0.738 |
| | (2.312) | (0.767) | (2.244) | (0.078) | (2.581) | (1.296) | (2.698) |
| RC | | | 0.958*** | | | | |
| | | | (0.210) | | | | |
| UR | | | | | -6.426*** | | |
| | | | | | (2.026) | | |
| ER | | | | | | | -0.436*** |
| | | | | | | | (0.142) |
| Control variables | YES | YES | YES | YES | YES | YES | YES |
| Area | YES | YES | YES | YES | YES | YES | YES |
| Year | YES | YES | YES | YES | YES | YES | YES |
| N | 209 | 209 | 209 | 209 | 209 | 209 | 209 |

Anhui. The western region consists of nine provinces, including Sichuan, Guizhou, Yunnan and Shaanxi.

The results for the Eastern Region are shown in Table 10. In the eastern region, the coefficient of direct influence of aging on carbon emission is not significant. This may be due to the fact that the eastern region is economically developed, has a more diversified and complex industrial structure, a larger scale of social production activities, and a higher labor participation rate. In addition, the degree of mechanization in the eastern region is generally high, and the demand for part of the working-age labor force is reduced through the substitution role of mechanical equipment. Therefore, the influence of aging on social and economic vitality is relatively limited. At the same time, the education level and quality of the population in the eastern region are relatively high, so the effect of aging to further promote the improvement of population quality to strengthen the low-carbon production and life style is not significant.

In the three intermediary paths of resident consumption, new-type urbanization and environmental regulation, the eastern region has passed the test significantly. In the eastern region, due to the high per capita income, the consumption level of residents is relatively higher, and the strong consumption demand may lead to the increase of production activities and the rise of carbon emission. With the change of population age structure, the carbon emission in this region can be reduced by restraining consumption to a certain extent. At the same time, due to the complex and diverse industrial structure and significant scale effect, the new urbanization rate driven by aging is conducive to further reducing regional carbon emission through population agglomeration and centralized utilization of energy [48]. In addition, due to the early start of the industry in the eastern region, the rules and systems are more sound. The improvement of the aging degree will further enhance the environmental regulation in the region through strengthening the environmental requirements, thus having a positive impact on reducing regional carbon emissions. To sum up, in the eastern region, the aging population mainly reduces the carbon emission of the region through three intermediary paths: inhibiting consumption, promoting urbanization and enhancing environmental regulation intensity.

Table 11 shows the results for the central region. In the central region, the deepening of aging is reflected in the increase of regional carbon emission. This is mainly because the central region is relatively late in industrial upgrading and transformation compared with the eastern

**Table 11. Heterogeneity in the central region.**

| variable | (1) | (2) | (3) | (4) | (5) | (6) | (7) |
|---|---|---|---|---|---|---|---|
| | CO2 | RC | CO2 | UR | CO2 | ER | CO2 |
| Main | | | | | | | |
| PA | 22.901*** | 1.428 | 22.915*** | -0.108 | 24.342*** | 5.843** | 25.833*** |
| | (8.453) | (0.962) | (8.395) | (0.150) | (8.434) | (2.666) | (8.772) |
| RC | | | -1.523** | | | | |
| | | | (0.751) | | | | |
| UR | | | | | 5.944 | | |
| | | | | | (4.561) | | |
| ER | | | | | | | -0.128 |
| | | | | | | | (0.276) |
| Control variables | YES | YES | YES | YES | YES | YES | YES |
| Area | YES | YES | YES | YES | YES | YES | YES |
| Year | YES | YES | YES | YES | YES | YES | YES |
| N | 152 | 152 | 152 | 152 | 152 | 152 | 152 |

region, and human capital is relatively insufficient. Under such circumstances, the aging of the population further aggravates the relative shortage of human capital in the region, which restricts the transformation and upgrading of local industries and the green and low-carbon development. Therefore, the appearance of "getting old before getting rich" makes the aging of the population in this region, on the contrary, increase the carbon emission of the region. At the same time, because this region is still in the early stage of new-type urbanization, the rapid expansion and construction of cities will produce a large amount of carbon emissions, and the development of regional industries is relatively insufficient, so the scale effect of carbon reduction brought by agglomeration has not been reflected. Therefore, in terms of intermediary effect, the new urbanization driven by aging will also increase the carbon emission of the region.

The heterogeneity of the western region can be obtained from Table 12. In the western region, due to its good ecological and environmental conditions and relatively low population

**Table 12. Heterogeneity in the western region.**

| variable | (1) | (2) | (3) | (4) | (5) | (6) | (7) |
|---|---|---|---|---|---|---|---|
| | CO2 | RC | CO2 | UR | CO2 | ER | CO2 |
| Main | | | | | | | |
| PA | -8.173** | 2.168* | -9.463* | 0.119 | -6.705 | -0.909 | -19.439*** |
| | (3.201) | (0.818) | (5.456) | (0.096) | (5.680) | (2.416) | (6.999) |
| RC | | | 1.249** | | | | |
| | | | (0.485) | | | | |
| UR | | | | | -8.414** | | |
| | | | | | (4.251) | | |
| ER | | | | | | | -1.132*** |
| | | | | | | | (0.211) |
| Control variables | YES | YES | YES | YES | YES | YES | YES |
| Area | YES | YES | YES | YES | YES | YES | YES |
| Year | YES | YES | YES | YES | YES | YES | YES |
| N | 209 | 209 | 209 | 209 | 209 | 209 | 209 |

density, the increase of the elderly population may bring new industrial opportunities, making the regional health care, culture and entertainment, tourism and other low-carbon service industries usher in great development, thus promoting the industrial structure of the western region to adjust to the direction of low-carbon economy. At the same time, because the western region is rich in natural resources and has a large number of renewable energy and clean energy, with the increase of the elderly population, people will pay more attention to health and environmental protection issues, so the government may increase investment in clean and renewable energy and reduce the use of traditional high-carbon energy. Thus, the decrease of regional carbon emission is promoted [49]. Therefore, in the western region, population aging can significantly reduce regional carbon emission.

## 7. Conclusions and policy implications

Based on the panel data of 30 provinces in China from 2004 to 2022, this paper uses three spatial weight matrices and a variety of model Settings to empirically analyze the spatial impact of population aging on regional carbon emission and its mechanism. The main conclusions are as follows:

The regional carbon emission showed significant spatial autocorrelation under the three spatial weight matrices of geographical adjacency matrix, geographical distance matrix and economic distance matrix. In addition, under the three spatial weight matrices, aging can significantly reduce the carbon emission of the region, while for the surrounding areas or areas with close economic ties, the industrial transfer brought by aging will increase the local carbon emission. From the time dynamic analysis, population aging has a promoting effect on regional carbon emission reduction, but with the passage of time, the promoting effect has weakened. The mediation effect model test shows that population aging has a complex mechanism on carbon emission. Population aging can reduce regional carbon emission by restraining household consumption, promoting new-type urbanization and enhancing environmental regulation intensity.

The influence of population aging on carbon emission exhibits pronounced regional disparities within China. Specifically, in the eastern and western regions, aging populations exert a negative impact on carbon emission, whereas in the central region, the effect is positive. These disparities are intricately linked to China's unique population migration dynamics.

With the acceleration of population mobility, the majority of migrants in China are young and middle-aged individuals. The eastern region attracts a significant influx of migrants, while the central region serves as the primary source of out migration, and the western region experiences relatively limited population mobility. Consequently, aging in the eastern and western regions primarily stems from declining birth rates, which often coincides with an abundant labor force that facilitates industrial restructuring and thereby contributes to the reduction of carbon emission. However, in the central region, the outflow of young and middle-aged workers results in what is termed "Hollowed-out aging." This demographic shift, characterized by a scarcity of youthful labor, poses challenges to industrial transformation and exacerbates carbon emission issues due to the central region's absorption of high-emission, energy-intensive industries transferred from the eastern region.

Moreover, the disparities in the impact of aging on carbon emission are further complicated by regional variations in economic development levels. In economically advanced eastern regions, aging populations are more likely to drive down carbon emission through mechanisms such as reduced consumption, accelerated urbanization processes, and stricter environmental regulations. Conversely, in relatively less developed western regions, aging populations contribute to lower carbon emission primarily by directly decreasing economic activity levels.

In addition to providing a series of empirical evidence for reducing regional carbon emission with aging, this paper also has the following policy implications:

Promote high-quality population development to reduce carbon emissions: To fully harness the potential of population, China must actively promote its high-quality development, shifting the focus of demographic dividends from mere quantitative growth to qualitative enhancement. This strategic transformation necessitates fostering a harmonious symbiosis between population growth and economic advancement, ensuring a continuous supply of talent that robustly supports economic restructuring, industrial upgrading, energy efficiency improvements, and ultimately, the deepening implementation of emission reduction efforts.

Orchestrating Regional and Cross-regional Industrial Harmony for Low-carbon Development: In the face of evolving and shifting local industries, adopting a balanced strategy is paramount. Within regions, tailored industrial policies should be formulated based on unique resource endowments and developmental needs, fostering green and low-carbon industrial growth patterns while optimizing industrial layouts and facilitating seamless industrial transfers. At the cross-regional level, enhanced cooperation and mutual support are essential to avert the "zero-sum game" of resource competition, jointly reducing the overall carbon footprint. Recognizing regional disparities in resource endowments and developmental stages, such collaboration holds profound significance for constructing a more sustainable economic landscape and achieving low-carbon development goals.

Promote population mobility and develop differentiated policies for carbon reduction in different regions: To promote population mobility and aggregation, deepen the process of new urbanization, China should ensure that the elderly enjoy a supportive living environment. China needs to integrate the power of the government, social institutions and families, strengthen the construction of the old-age security system, and accelerate the improvement of rural elderly care infrastructure to reduce the burden of family care. Given the regional differences in the impact of aging on emissions, it is crucial to formulate differentiated policies. In the eastern region, it is necessary to focus on cultivating low-carbon consumption habits and promote the process of urbanization on the basis of strengthening environmental protection; The central region should prioritize attracting talents, accumulating human capital, and closely integrating the process of urbanization with the concept of green development; The western region needs to rely on the emerging low-carbon service industry to promote the transformation of energy structure and achieve the goal of regional low-carbon development.

## Author Contributions

**Conceptualization:** Xueqiang Ji, Yuesong Zhang.

**Formal analysis:** Xueqiang Ji.

**Funding acquisition:** Yuesong Zhang.

**Investigation:** Haohao Li.

**Methodology:** Haohao Li, Xueqiang Ji.

**Project administration:** Xueqiang Ji.

**Software:** Zhuoqun Li.

**Supervision:** Yuesong Zhang.

**Visualization:** Zhuoqun Li.

**Writing – original draft:** Zhuoqun Li, Haohao Li.

**Writing – review & editing:** Zhuoqun Li.

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
