## [Decision Letter · Decision Letter 0]

15 May 2024

PONE-D-24-08676Study on the spatio-temporal impact effect and impact path of population aging on carbon emission intensityPLOS ONE

Dear Dr. LI,

Thank you for submitting your manuscript to PLOS ONE. After careful consideration, we feel that it has merit but does not fully meet PLOS ONE’s publication criteria as it currently stands. Therefore, we invite you to submit a revised version of the manuscript that addresses the points raised during the review process.

We look forward to receiving your revised manuscript.

Kind regards,

Alejandro Botero Carvajal, MD

Academic Editor

PLOS ONE

Journal Requirements:

"This work was Supported by the National Natural Science Foundation of China under Grant No. 72274207, "Research on Cross-domain Governance Models of Beijing-Tianjin-Hebei Housing Market in the Context of Regional Synergistic Development"."

3. We note that your Data Availability Statement is currently as follows: All relevant data are within the manuscript and its Supporting Information files

**Additional Editor Comments:**

 Please ensure to adress these points:1. The Topic some way has confusion on term " impact effect" please clarify 

2. The findings on Eastern and Western different from central part of China shows difference while conclusion remains the same, please explain in the discussion section.

3. Spatial information better to be displayed with GIS supported maps. 

4. Variation across temporal period not well discussed on displayed analysis table. Please clarify. 

5. Other factors related to CEI not mentioned in related to age. Please clarify.

6. Different sources of data indicated, but specifically which data from where not mentioned. Please clarify. 

Reviewers' comments:

Reviewer's Responses to Questions

**Comments to the Author**

1. Is the manuscript technically sound, and do the data support the conclusions?

Reviewer #1: Partly

2. Has the statistical analysis been performed appropriately and rigorously? 

Reviewer #1: Yes

3. Have the authors made all data underlying the findings in their manuscript fully available?

Reviewer #1: Yes

4. Is the manuscript presented in an intelligible fashion and written in standard English?

Reviewer #1: Yes

5. Review Comments to the Author

Reviewer #1: 1. The Topic some way has confusion on term " impact effect"

2. The findings on Eastern and Western different from central part of China shows difference while conclusion remains the same

3. Spatial information better to be displayed with GIS supported maps

4. Variation across temporal period not well discussed on displayed analysis table

5. Other factors related to CEI not mentioned in related to age

6. Different sources of data indicated, but specifically which data from where not mentioned

6. PLOS authors have the option to publish the peer review history of their article (what does this mean?). If published, this will include your full peer review and any attached files.

Reviewer #1: **Yes: **Tofik Abajebal

---

## [Author Response · Author response to Decision Letter 0]

26 Jul 2024

Response to Reviewers

Dear Editor and Reviewers,

Re：Study on the spatial-temporal impact effect and impact path of population aging on carbon emission intensity [PONE-D-24-08676] - [EMID: 4ad6c19525c22538]

Thank you for your letter and for the reviewers' comments regarding our manuscript entitled "Study on the spatial-temporal impact effect and impact path of population aging on carbon emission intensity". Those comments are very valuable and have greatly contributed to improving the quality of our work.we have carefully considered each of the comments and have made revisions accordingly. Below are the specific details of the modifications I have made.

Journal Requirements

1.Please ensure that your manuscript meets PLOS ONE's style requirements, including those for file naming. 

Response: Thank you for your thorough review and valuable feedback. We have carefully revised our manuscript to ensure that it fully meets PLOS ONE's style requirements, including those for file naming. We have followed the formatting guidelines provided at the links you shared, and renamed all files according to PLOS ONE's conventions. We appreciate your time and look forward to your further assessment.

2.Please state what role the funders took in the study.  

Response:Thank you for bringing up the importance of clarifying the funder's role in our study. We would like to confirm that the funders, the National Natural Science Foundation of China under Grant No. 72274207, had no role in the study design, data collection and analysis, decision to publish, or preparation of the manuscript. This statement has been included in our revised manuscript and will also be reflected in our cover letter. We appreciate your attention to this matter and look forward to your further review.(page 36,line 809-813)

3. We note that your Data Availability Statement is currently as follows: All relevant data are within the manuscript and its Supporting Information files

Response: Thank you for your review and attention to our Data Availability Statement. Upon further review, we confirm that the data in the article are indeed sourced from third parties, specifically the Office for National Bureau of Statistics of China (http://www.stats.gov.cn/). We have clarified in the manuscript that all relevant data, as specified and used in our study, can be accessed through this website, with some potentially available for a fee. We understand that PLOS requires the minimal data set to be available for replication of study findings, and we have ensured that the data accessed and utilized in our work meet this criterion to the best of our abilities, given the constraints of third-party data access. We will update our Data Availability Statement as below accordingly to reflect this information.

“The data in the article are all from third parties, as specified in the article, and all of these data can be accessed through the website, some of which can be obtained for a fee. All the data in the article comes from the Office for National Bureau of Statistics of China (http://www.stats.gov.cn/).”

4. Please review your reference list to ensure that it is complete and correct.

Response: Thank you for bringing this to our attention. We have thoroughly reviewed and updated our reference list to ensure it is complete, accurate, and free of retracted articles. Any retracted papers have been removed and replaced with appropriate current references. We have noted any changes made in the revised manuscript.

Next is our response to the reviewers' comments:

Reviewer #1

1. The Topic some way has confusion on term " impact effect" please clarify. 

Response: Thank you for your comments. We've revised the title to "Shaping a Low-Carbon Future: Uncovering the Spatial-temporal Effect of Population Aging on Carbon Emissions in China" to avoid confusion and clarify the focus. Please find the revised manuscript attached.（page1, line 4-5）

2. The findings on Eastern and Western different from central part of China shows difference while conclusion remains the same, please explain in the discussion section.

Response: Thank you for your valuable comments. We fully agree with your observation that our previous discussion on the different findings between eastern, central, and western China was insufficient. To address this, we have made corresponding supplements and improvements in the discussion section. The following content we added aims to more clearly explain the reasons for the differences in findings and the heterogeneous conclusions among the eastern, central, and western regions(page34, line 663-682):

“The influence of population aging on carbon emission exhibits pronounced regional disparities within China. Specifically, in the eastern and western regions, aging populations exert a negative impact on carbon emission, whereas in the central region, the effect is positive. These disparities are intricately linked to China's unique population migration dynamics.

With the acceleration of population mobility, the majority of migrants in China are young and middle-aged individuals. The eastern region attracts a significant influx of migrants, while the central region serves as the primary source of out migration, and the western region experiences relatively limited population mobility. Consequently, aging in the eastern and western regions primarily stems from declining birth rates, which often coincides with an abundant labor force that facilitates industrial restructuring and thereby contributes to the reduction of carbon emission. However, in the central region, the outflow of young and middle-aged workers results in what is termed "Kongxin aging." This demographic shift, characterized by a scarcity of youthful labor, poses challenges to industrial transformation and exacerbates carbon emission issues due to the central region's absorption of high-emission, energy-intensive industries transferred from the eastern region.

Moreover, the disparities in the impact of aging on carbon emission are further complicated by regional variations in economic development levels. In economically advanced eastern regions, aging populations are more likely to drive down carbon emission through mechanisms such as reduced consumption, accelerated urbanization processes, and stricter environmental regulations. Conversely, in relatively less developed western regions, aging populations contribute to lower carbon emission primarily by directly decreasing economic activity levels..”

3. Spatial information better to be displayed with GIS supported maps. 

Response: Thank you for your valuable comments on our manuscript. We appreciate the suggestion to enhance the spatial information presentation through GIS-supported maps.We have carefully considered your feedback and made the following revisions to better illustrate Based on the index system constructed in this paper, to enhance the intuitiveness of the results, we selected three time points: 2004, 2012, and 2022. The breakpoint method was employed to classify both carbon emission and the degree of aging into five distinct levels. Subsequently, ArcGIS software was utilized to generate distribution maps illustrating the carbon emission and aging degrees.

Fig 2. Spatial Distribution of carbon emission and aging in China, 2004, 2012 and 2022

“As depicted in Fig.2, during the sample period, China's carbon emission exhibited an overall steady decline. Notably, the carbon emission displayed geographical disparities, with higher intensities prevalent in resource-rich provinces in the western and northeastern regions, and comparatively lower intensities observed in the eastern coastal provinces. Additionally, nationwide, carbon emission significantly decreased over time, except for resource-based provinces in the central and western parts of the country, including Inner Mongolia and Shanxi. By 2022, the carbon emission of major provinces in southern China had almost entirely shifted to the lowest range.China's population aging is gradually deepening, with higher levels of aging geographically concentrated in some of the eastern coastal and southern provinces. Except for a few western provinces such as Xinjiang and Qinghai, there is a clear trend of deepening aging throughout the country. In 2004, the deeply aging provinces included only a few provinces such as Jiangsu, Beijing, Tianjin, and Chongqing, but by 2012, the range had expanded significantly, with many eastern and central-western provinces such as Shandong, Anhui, Sichuan, and Hunan entering the deeply aging range one after another. By 2022, the range has further expanded, and the level of aging has risen significantly across the Northeast region, with Liaoning Province also entering the deep aging zone.”

4. Variation across temporal period not well discussed on displayed analysis table. Please clarify. 

Response: Thank you for your valuable suggestions. Tables 6, 7, and 8 in the paper utilize rolling window analysis based on different forms of geographical matrices to reveal the temporal variations in the impact of aging on carbon emission. The results indicate that the interaction term between aging and time is positive throughout the entire period, suggesting that the emission-reducing effect of aging on carbon emission is gradually declining over time. To further explain this phenomenon, we have updated certain sections of the article as follows(page25, line 492-505):

“Table 6 shows the change of the influence of aging on carbon emission under the geographical adjacency matrix under the rolling time window. It can be found that the coefficient of aging has been significantly negative in the whole time interval, indicating that the overall aging can still effectively reduce the regional carbon emission. However, in the whole time interval, the interaction term with time is positive, indicating that the emission reduction effect of aging on carbon emission is gradually declining with the passage of time. This phenomenon may be due to a combination of factors. On the one hand, economic and technological developments may make other factors, such as the adoption of cleaner energy sources and improvements in energy efficiency, more important drivers of carbon emission, thus reducing the direct effect of ageing.On the other hand, the effect of ageing on reducing carbon emission may also be offset to some extent by higher incomes and higher consumption levels of the older population over time.In addition, changes in the policy environment, such as the introduction of energy-saving and emission-reduction policies and an increase in people's awareness of environmental protection, may offset to some extent the reducing effect of ageing on carbon emission.”

5. Other factors related to carbon emission not mentioned in related to age. Please clarify.

Response: Thank you for your insightful comments, which are of great significance in enhancing the completeness of the article. We have supplemented the paper by including the impact of other factors on carbon emission both within the province and in neighboring provinces. The specific additions are as follows(page20, line 434-450):

“In addition to the degree of ageing, we also consider other factors that may potentially influence the carbon emission within the province as well as in neighboring provinces.The level of local economic development (lnGDP) has no significant impact on the province's carbon emission, yet it significantly reduces the carbon emission of neighboring provinces. This phenomenon can be attributed to the effects of technology transfer and industrial cooperation. The degree of openness (lnOPEN) and urban construction (UC) do not significantly influence carbon emission. The enhancement of science and education (lnEDU) has a pronounced negative effect on the province's carbon emission, suggesting that technological innovation and educational advancement contribute to reducing carbon emissions within the region. However, it shows a positive effect in neighboring regions, which may mirror the competition for resources and the environmental pressures among these regions. Government intervention (GI) has a non-significant impact on the province's carbon emission but exerts a significant negative effect on the carbon emission of neighboring provinces, highlighting the government's positive role in fostering regional environmental cooperation and mitigating carbon emissions. Economic structure (ES) primarily exerts a negative influence on the province's carbon emission, while their effect on neighboring regions is insignificant. This could be linked to the fact that a consumption-driven economy diminishes the province's reliance on heavy industry and energy-intensive sectors..”

6. Different sources of data indicated, but specifically which data from where not mentioned. Please clarify. 

Response: Thank you for bringing this oversight to my attention. In my manuscript, I acknowledged the various data sources utilized for the analysis, yet I acknowledge the oversight of failing to furnish explicit details on the precise datasets and their respective origins. I sincerely apologize for any ambiguity or confusion this may have inadvertently caused. To rectify this, I have now included a detailed clarification on the origins of the various data sources utilized within the paper, as follows(page12, line 297-305):

“The data used in this paper were from "China Statistical Yearbook", "China Energy Statistical Yearbook" and "China Environmental Statistical Yearbook". Specifically, data related to carbon emission come from the "China Energy Statistics Yearbook", data related to environmental regulation on three-waste emissions come from the "China Environment Statistics Yearbook", and all other variables come from the" China Statistics Yearbook "of previous years. Considering the availability and completeness of data, Hong Kong, Macao, Taiwan and Tibet Autonomous Region of China were deleted, and the interpolation method was used to process some missing data. Finally, we select the panel data of 30 provinces in China from 2004 to 2022 as the basis of the study. The descriptive statistical results of each variable are shown in Table 1.”

The following are additional modifications made the author team.

1.Data updating and result robustness verification:

The author team obtained the latest dataset covering the period from 2004 to 2022. This data update ensures the timeliness and representativeness of the analysis results. Through detailed data processing and analysis, we were pleasantly surprised to find that despite the significant expansion of the data range, the results were basically consistent with previous research findings. This finding not only highlights the robustness of the conclusions of this study, that is, the same trend or pattern can be consistently observed in different time periods, but also further consolidates the theoretical basis and practical application value of the research. It means that the phenomenon or mechanism revealed in this study may have a broader universality and profound impact, providing solid support for future research in related fields.

2.Writing style optimization and summary introduction improvement:

In view of the excellent reputation of PLOS ONE as an open access scientific journal and its unique writing style guidelines, the author team conducted in-depth research on journal norms and requirements, and used this as a blueprint to comprehensively and carefully revise and improve the abstract and introduction sections of the paper. In the abstract section, we strive to concisely summarize the research background, purpose, methods, main findings, and conclusions, while ensuring that the language is concise and the logic is clear, so that readers can quickly grasp the core value of the paper. In addition, we also pay special attention to the objectivity, accuracy, and completeness of the abstract, avoiding any expressions that may cause misunderstandings.

For the introduction, we have focused on strengthening the depth and breadth of the literature review, systematically reviewing the research progress in related fields at home and abroad, and clarifying the position and contribution of this study in the academic system. At the same time, we h

---

## [Editor Report · Decision Letter 1]

6 Aug 2024

Shaping a Low-Carbon Future: Uncovering the Spatial-temporal Effect of Population Aging on Carbon Emissions in China

PONE-D-24-08676R1

Dear Dr. 李,

We’re pleased to inform you that your manuscript has been judged scientifically suitable for publication and will be formally accepted for publication once it meets all outstanding technical requirements.

Kind regards,

Alejandro Botero Carvajal, MD

Academic Editor

PLOS ONE

---

## [Editor Report · Acceptance letter]

16 Sep 2024

PONE-D-24-08676R1 

PLOS ONE

Dear Dr. LI, 

I'm pleased to inform you that your manuscript has been deemed suitable for publication in PLOS ONE. Congratulations! Your manuscript is now being handed over to our production team.

Kind regards, 

on behalf of

Dr. Alejandro Botero Carvajal 

Academic Editor

PLOS ONE